# Evaluation of the Fatigue Behaviour and Failure Mechanisms of 52100 Steel Coated with WIP-C1 (Ni/CrC) by Cold Spray

**DOI:** 10.3390/ma15103609

**Published:** 2022-05-18

**Authors:** Viorel Goanta, Corneliu Munteanu, Sinan Müftü, Bogdan Istrate, Patricia Schwartz, Samuel Boese, Gehn Ferguson, Ciprian Ionut Morăraș

**Affiliations:** 1Mechanical Engineering, Mechatronics and Robotics Department, Mechanical Engineering Faculty, “Gheorghe Asachi” Technical University of Iasi, 700050 Iasi, Romania; viorel.goanta@academic.tuiasi.ro (V.G.); ciprian-ionut.moraras@academic.tuiasi.ro (C.I.M.); 2Technical Sciences Academy of Romania, 26 Dacia Blvd., 030167 Bucharest, Romania; 3Department of Mechanical and Industrial Engineering, Northeastern University, Boston, MA 02115, USA; s.boese@northeastern.edu; 4Kostas Research Institute, Northeastern University, Burlington, MA 01803, USA; t.schwartz@northeastern.edu; 5Army Research Laboratories, Aberdeen Proving Ground, Maryland, MD 21005, USA; gehn.ferguson@jhuapl.edu

**Keywords:** coating, fatigue, microscopic observations, fissure, crack initiation, fatigue limit

## Abstract

Cold spray technique has been major improved in the last decades, for studying new properties for metals and alloys of aluminum, copper, nickel, and titanium, as well as steels, stainless steel and other types of alloys. Cold sprayed Ni/CrC coatings have the potential to provide a barrier as well as improved protection to steels. Fatigue characteristics of 52100 steel coated with Ni/Chrome-Carbide (Ni/CrC) powder mixture by using cold gas dynamic spray are investigated. Fatigue samples were subjected to symmetrically alternating, axially applied cyclic fatigue loading until failure. The test was stopped if a sample survived more than 5 × 10^6^ cycles at the applied stress. Fracture surfaces for each sample were examined to investigate the behaviour of the coating both at high stress levels and at a high number of stress cycles. Scanning electron microscopy was used to assess the damage in the interface of the two materials. Good fatigue behaviour of the coating material was observed, especially at low stresses and a high number of cycles. Details of the crack initiation region, the stable crack propagation region and the sudden crack expansion region are identified for each sample. In most of the samples, the initiation of the crack occurred on the surface of the base material and propagated into the coating material. The possible effects of coatings on the initial deterioration of the base material and the reduction of the lifespan of the coated system were also investigated. The aim of the paper was to study the interface between the base material and the coating material at the fatigue analysis for different stresses, highlighting the appearance of cracks and the number of breaking cycles required for each sample.

## 1. Introduction

Coatings are used to enhance surface properties or protect the base material from degradation caused by the interaction with the external environment such as corrosion, attack of various chemicals, erosion, effects of high temperature or other external aggressions such as the high-velocity impact of particles/components. The researchers must also consider the compatibility between the base material and coating material. In addition to many other properties that need to be considered, obtaining the mechanical properties of the coating itself must also be considered. Depending on the chemical composition and coating process used, the coating and base materials may differ from each other in terms of physico-mechanical properties [1]. Major differences between such properties of coating materials and base materials include thermal expansion, specific deformation, yield limit, fatigue behaviour and creep [2,3]. Important differences in properties can cause damage to the interface between the coating and the base material and failure of the structure of the coating material. If significant differences occur between the properties of the two materials, it can lead to a significant reduction in the lifespan of the assembly, basic material—coating material. As a result, the mechanical properties of the coating systems, as well as the prevailing mechanisms of damage will be determined by the critical property relevant to the entire assembly, basic material—the coating material. In case the thickness of the coating is much lower than that of the base material, the critical property is the difference between the specific deformations of the two materials for dynamic loading [4]. Surface temperature variations may result in changes in the mechanical and physical properties of the coating. In addition, micro-defects due to effects such as stress corrosion cracking can also occur. It is also possible for the coating process, depending on the method used, to damage the surface of the base material and/or to introduce local residual stress that is of the same sign as those arising from the mechanical loading [5,6,7,8,9]. This may cause premature destruction of a coated component. It is well known that surface quality plays an important role in the fatigue life of a component subjected to dynamic loading. A coating may have a beneficial role in terms of protection against environmental effects, but other operating conditions such as the fatigue life of the component must not be adversely affected. Any coating system that improves certain properties relative to the uncoated material, may shorten the fatigue life of a component due to cracks that begin in the coating and propagate in the substrate material [5,6]. In this context, research has focused on investigating the effect of coatings on the fatigue behavior of the resulting components [10,11,12,13,14,15,16]. This work showed different opinions regarding the mechanical characteristics, especially the fatigue analysis, following the coating process [17]. It is reasonable to expect that the change of mechanical properties depends on a number of factors that need to be investigated separately for each case.

Cold spray is a particle deposition technology in which micron sized particles are accelerated through a converging-diverging de Laval nozzle to velocities that can reach supersonic velocities [18]. Upon impact, the particles undergo severe plastic deformation and adhere to the substrate through a combination of metallic bonding and mechanical interlocking. The particles gain momentum and some heat from the hot gas flowing through the nozzle but remain below the melting temperature of the material. Thus, most of the material properties of the powder are preserved in the coating, and cold spray can be considered a solid-state material deposition technology. A wide range of ductile metal powders has been shown to work well with this technology when deposited on ductile metal surfaces. Ceramic powders, which tend to be good insulators and have good wear resistance, can be successfully deposited by using a ductile material as the matrix [19]. Cold spray metal matrix composite coatings have even been used as a coating to improve the fatigue life of additive manufactured components [20].

The present work aims to assess the effect of coating 52100 steel with a 0.5 mm thick Ni/CrC mixture by using cold spray technology on the fatigue life of the combined material system. To this end alternating—symmetrical axial stresses are applied on coated, cylindrical specimens. This subjects the sample to compression and tensile cycles. The tensile side of the cycle is useful to study the endurance of the coating. The cylindrical shape ensures that the stresses introduced into the sample are uniform throughout the cross-section of the sample. Thus, any defects that will lead to the onset of fatigue can be highlighted. The mechanical behaviour of the coated material system, the coating, the interface area, the crack initiation area and the regions of stable or unstable crack propagation are investigated. The paper also compared the fatigue resistance of 52100 steel given in reference [21] and the behaviour of the coating assembly presented in this paper, respectively, coating 52100 steel with a Ni/CrC mixture by using the cold spray technology.

## 2. Materials and Methods

The morphological aspect of the surfaces was determined using the SEM Quanta 200 3D (FEI, Brno, Czech Republic) electron microscope with LFD (Large Field Detector), a High Voltage of 20 kV, a spot size of 5 and a working distance of 15 mm. The X-ray diffractions (XRD, Panalytical, Almelo, The Netherlands) were performed using an Xpert PRO MPD 3060 facility from Panalytical (Almelo, The Netherlands), with a Cu X-ray tube (Kα = 1.54051 Å), 2 theta: 10°–90°, step size: 0.13, time/step: 51 s, and a scan speed of 0.065651°/s.

Figure 1 presents the diffractograms for the base and coating material. Knowing the chemical composition of the powder [20], the predominant phase of NiCr (cubic structure -ICDD 96-901-2971) was obtained at a 2 Theta angle of approximately 44° and the secondary phase of Cr_3_C_2_ (cubic structure-ICDD 96-901-1599) at a 2 Theta angle of approximately 53° and 77°. In the case of the basic material, the predominant phase is Fe (cubic structure -ICDD 96-901-6602), around the 2 Theta angle of 45°, and the secondary phases contain Fe and Cr3C2 at the 2 Theta angles of 65° and 83°.

The dimensions of the cylindrical specimens for testing the fatigue characteristics of the coated 52100-steel were determined according to the ASTM E4666-15 standard (Figure 2a). The specimens were coated with a mixture of Chrome-Carbide and Ni powder by using a VRC, Gen-III cold spray machine (VRC Metal Systems, LLC, South Dakota, USA). The parameters used for spraying the powder mixture are given in Table 1. The powder mixture is commercially known as WIP-C1 [22]. Before the surface is coated with WIP-C1 powder a thin layer of bond coat that consists of WIP-BC1 powder was applied on the surface with a 60° nozzle orientation. This operation has multiple effects to shot-peen, clean and roughed the surface, in addition to creating a thin layer of WIP-BC1 coating. Additionally, the bond layer improves the adhesion of the WIP-C1 layer. The cylindrical specimens were mounted on a lathe and the WIP-C1 coating was applied by moving the nozzle axially, while holding it normal to the specimen axis. The spray conditions used to deposit the coatings are given in Table 1 and in references [20,23].

A standard tensile test (ASTM E8/E8M-16) was used to determine the material response of the coated system. This test was carried out on three coated samples, to ensure that the material system used in cyclic fatigue tests is well characterized. The specimen dimensions and a coated specimen are shown in Figure 2a,b. A typical stress-strain curve of the coated system is shown in Figure 3. The initial yield stress of the material before the application of cyclic loading is determined from these tests among other material characteristics. These include the stress level at which compatibility issues between the strains of the base and coating material become significant. To this end, the surface of the coated samples was monitored with a video camera during the test to look for the initiation of possible damage in the form of surface cracks.

The static loading was performed on a universal testing machine, Instron 8801 (Instron, Norwood, MA, USA) (Figure 2c). An extensometer was mounted in order to measure strain accurately (Figure 2c). Since the total deformation of the sample was not very large, it was possible to acquire data with the extensometer until the sample broke. A video of the calibration area was made for capturing the moment of the appearance of the crack in the coating material (Figure 4a). The point on the stress-strain curve indicating the moment of the appearance of the crack was found by overlapping the time signatures of the video and that of the extensometer data. A broken coated sample can be seen in Figure 4b.

Both from the examination of the fractured sample (Figure 4b) and from the stress-strain curve given in Figure 3 the following observations can made: (i) the shape of the stress-strain curve is dominated by the behavior of the base material, 52100-steel); (ii) the elastic region is relatively wide; high values of the load are reached on a linear path and the yield point is close to the maximum stress; (iii) a material hardening region is not observed; after reaching the offset yield point, Rp_0.2_ or σ_p0.2_, the material undergoes a pronounced elongation, but without further increase of stress; (iv) on the stress-strain curve, the cracking point of the coating material was marked, which was determined by the study of the videos captured during the test, as explained above; this point corresponds to a strain of 2.64%; (v) cracking of the coating material does not occur in the elastic region of the stress-strain curve; (vi) the strain at which the cracking of the coating material takes place can be obtained locally, where a microcrack is initiated in the base material and where, under the conditions of existence of a stress concentrator (crack for example), the local deformations become large; (vii) given that the total strain is 14%, the base material undergoes significant plastic deformation, hence the contraction is observed in Figure 4b; (viii) the coating material suffers significant damage; (ix) it is found that the σp_0.2_ is 831 MPa, a value from which it starts to decrease in fatigue loading.

## 3. Results and Discussion

The ASTM E466-15 standard is used for fatigue testing. The shape and dimensions of the samples are shown in Figure 2a. The fatigue tests were conducted after a symmetrical, alternating cycle. For the fatigue testing, the initial load must be in the elastic region (Figure 3). The fatigue test is carried out on nine samples, starting with a load value corresponding with the stress close to the yield limit and continuing by decreasing the load. The first maximum stress was 76.5% of the yield limit value σ_max_ = 0.765, σ_p0.2_ = 636 MPa. Table 2 shows the maximum stress values used in the tests.

The fracture surfaces were investigated at macro- and micro-scales by using optical and scanning electron microscopy. Observations of fracture of the coating material in different areas of the surfaces are presented. It is expected that the base and the coating materials will experience different strain levels where ε_base_ > ε_coating_. Significant damage to the coating material could be expected to occur, given that the two materials deform differently, and large deformations occur at higher values of the stress cycles. In order to be able to observe this, images were taken from the fatigue test, from the central and calibration areas of the samples. From the SEM images (Figure 5) the grip of the coating material on the base material was a good one and there were no significant differences in the fatigue behaviour of the coating and base material.

As can be seen in Figure 6, no damage to the coating material was observed even at more than five million cycles, nearly up until fracture. The macroscopic image analysis shows a very good behaviour of the coating material during the test where the material experienced both tensile and compressive strains.

The moment of propagation of the fatigue crack is captured in Figure 7. It is known that, depending on the material, a relatively large number of fatigue cycles may be required to initiate a crack. After initiating the crack, depending also on the value of the tensile stress, the number of cycles for fatigue crack propagation may vary. In Figure 7, the visible crack (below the hand drawn line) is the one propagated by fatigue; a relatively small number of cycles were required until the final, sudden breakage of the sample soon after this crack appeared. From this figure, it can be observed a macroscopic propagation of the crack and the sample is close to the final breakage. No detachment of the coating material, near the front of the propagated crack, is seen. As a result, the propagation of the crack in the coating material follows the same path as in the base material, without any collateral damage to the coating material. From a macroscopic point of view, the coating behaves well during cyclic tests with alternating, symmetrical, axial loading.

As mentioned earlier, the tests were conducted from higher stress values (resulting in lower values of the number of stress cycles) to lower stress values. The stress cycle values of 5 million have been set as a limit. If the sample did not brake up to this number of stress cycles, the test was stopped, regardless of the stress values. Table 2 shows the number of cycles to failure for each maximum stress level.

For the fatigue analysis, 9 samples of 52100 steel were used as the base material, (Figure 8), coated using the method presented above. Samples 1 and 8 did not break even after 5 million cycles, as a result, the test was stopped, and the samples remained unbroken. The other samples fractured at different values of loads and number of cycles. In what follows observations about each sample are presented in detail.

### 3.1. Samples 1 and 8 (σ_max_ (MPa); N), (420; 5,519,600) and (438; 5,451,948)

Samples 1 and 8 did not break even after more than five million cycles of stress reversals. Examining the outer surface of the sample, no damage to the coating material was found (Figure 9). For Sample 1, the coefficient resulting from the ratio of the test loads to the yield limit is: K_σF_ = σ_test_/σ_y_ = 438/831 = 0.53. As a result, we can state that when the loading is at 53% of the yield limit, and if the sample doesn’t fracture, the coating material withstands high values of the number of cycles, not causing significant damage to the surface. This statement must be taken with caution, since it is not known what happens at the interface between the two materials, in the area where the first plastic deformations of the base material occur. At low loads, in the elastic region, the deformations of the base material are not very different from the deformations of the coating material [19]. Under these conditions, the tension and compression of the coating materials are similar to the expansion and compression of the base material. Although the stress is applied over 5 million cycles, the loads being low, the coating material withstands very well within the stresses of cyclic fatigue, without experiencing significant damage.

The macroscopic and microscopic observations were made on the fracture surfaces, of each failed sample separately. The crack initiation and the behaviour of the coating material are presented in descending order starting with the maximum load.

### 3.2. Sample 2 (σ_max_ (MPa), N), (636, 5300)

Macroscopic observations: Longitudinal cracks were observed in the coating material when the sample was subjected to high stresses, close to the flow limit (Figure 10a). In the area of the crack initiation, the surface of the coating material is flat, without obvious macroscopic cracks in the plane of fracture. Throughout the crack, the surface of the coating material remains flat, there being no detachment from the material (Figure 10b). At the boundary between the areas of the slowly propagating crack and suddenly propagating crack due to fatigue, significant damage to the coating material is seen (Figure 10c). Longitudinal cracks that are not seen in the other samples were found here. The longitudinal cracks appear only in the area of the suddenly propagating crack region. In the area of stable propagation of the crack front, no such cracks are found.

Microscopic observations: In the area of crack initiation, there is no significant damage to the coating (Figure 10d), like a macroscopic detachment from the material, cracks inside the coating material, and cracks in the particles that are in the coating material. In the zone of the crack initiation (Figure 10e), deterioration can be found in the area of the coating material with microscopic detachment from the material. In this area, no damage to the coating material is found on the outside of the sample and it can be concluded that the crack was initiated, first in the base material, after which it extended in both directions, both in the coating material, outwards and in the base material, inwards. In Figure 10e detachment of the coating material from the base material is observed. This occurred as a result of the differences between the Young modulus of the two materials, thus of the differences in specific deformation, emphasized earlier in the paper. As a result, in the area of crack initiation, the local strain of steel reached the value of 2.64%.

### 3.3. Samples 3 and 4, (σ_max_ (MPa), N), (530, 34,660) and (477, 878,965)

Macroscopic observations: Figure 11a–c show that at the end of the crack propagation period, the fracture surface changes direction, from being perpendicular to the force direction, to an orientation of approximately 45°. This can be explained as follows. While moving in the base material, the crack front encounters resistance to advancing in the initial direction; given that a large part of the surface has been broken, the outer force, which introduces high loads, acts unbalanced in relation to the remaining fracture surface (Figure 11f). Given the high loads in the remaining uncracked surface, the movement of the dislocations in the base material, under the action of tangential loads, is done in an accelerated manner. In the region where the front of the propagated crack changes direction, significant detachment of the coating material from the base material is found (Figure 11c).

Microscopic observations: In the final fractured surface, large damage to the coating material is noted (Figure 11c). For all samples, the large damage to the coating material in the final fracture surface is due to the large deformation differences between the base and coating material. The base material having a high deformation (in the plastic domain) in the final fracture surface, also trains the coating material. Thus, the coating, not being able to deform in the same way as the base material, deteriorates by the appearance of large cracks. In the area of crack initiation (Figure 11d,e), there is no significant damage to the coating material. Detachment of the coating material from the base material due to the deformation differences between the two materials is observed.

### 3.4. Samples 4 and 7, (σ_max_ (MPa), N), (477, 878,965) and (459, 2,287,070)

Macroscopic observations: For Samples 4 and 7 the conclusions are similar. The loads used for Samples 4 and 7 are increasingly low. This results in an increasingly large number of fatigue cycles before a fracture occurs. The fracture occurs by stable propagation of the crack through fatigue and no deflection of the crack is observed (Figure 12a–c). However, in these samples, the load is higher than in the following samples and the area of stable crack propagation is smaller (see the corresponding figures in the following samples). No damage is observed due to the removal/detachment of particles or by the appearance of large cracks, in the final fracture surface (Figure 12c).

Microscopic observations: In the area of crack initiation by fatigue, no significant damage to the coating material is found (Figure 12c). At higher magnifications, in the area of crack initiation, a detachment is revealed by showing a radial crack through the coating material (Figure 12d,e).

### 3.5. Samples 5 and 9, (σ_max_ (MPa), N), (449, 3,456,873) and (446, 4,424,120)

Macroscopic observations: For Samples 5 and 9 the conclusions are similar. These samples were tested at low loads, and more than 50% of the fracture surface is obtained by stable propagation of the crack. The coating uniformly broke throughout its circumference as shown in Figure 13a,b. A detachment of the coating material from the base material is found at the beginning of the sudden propagation of the crack front (Figure 13c).

Microscopic observations: In the area of crack initiation by fatigue, no significant damage to the coating material is observed (Figure 13d,e). A very fine crack between the base material and the coating material meaning a detachment between the two materials appears in the area of initiation of the fatigue crack. In this sample, inter- and intra-granular microcracks (Figure 13e), are observed more pronounced than in the previous samples, especially near the interface area between the two materials.

The data presented in Table 2 is plotted in a Wöhler diagram in Figure 14. Sample 8 (438 MPa) and Sample 1 (420 MPa) did not break after 5 million cycles. The appearance of the diagram is a conventional one, with a fatigue limit of around 446 MPa, given the imposed limit of 5 million cycles.

Tests performed by other researchers on the uncoated 52100 material have shown that no limit to well-defined fatigue can be determined [20]. On the other hand, the base material, in this case, 52100 steel significantly influences the appearance of the S-N durability curve.

## 4. Conclusions

The coated parts will have to work in the area of the high durability of Wöhler’s curve meaning at relatively low stress levels. If parts are subjected to high stresses, significant damage to the coating material may occur, caused by differences in deformation, in the local plastic domain, between the base material and the coating material. The behaviour in the area of crack initiation is of particular interest, due to the greatest number of stress cycles used to initiate the crack and, by design, we do not have to reach the area of sudden propagation of the crack. An area of interest is also the stable propagation of the fatigue crack, where, if the crack is observed, early measures can be taken to replace/repair the cracked part.

The microscopic study of the surfaces resulting from the fatigue fracture shows the following:-The two areas of the same surface are highlighted, an area characteristic of the propagation by fatigue in a long time and an area characteristic of the sudden crack propagation, until the final fracture;-The area of stable fracture propagation by fatigue by finer grains; the area from the sudden breakage is characterized by coarser grains;-In the crack initiation area, there are no pores or major cracks between the coating material and the base material;-The initiation of the crack took place on the surface of the base material, then extended to the coating material and inside the base material;-Crack initiation taking place on the surface of the base material suggests that applying the coating is unlikely to negatively impact the fatigue properties of the base material;-The coating material did not suffer detachment at the beginning of fatigue cracking;-In the whole area where the crack was propagated by stable fatigue (lower stresses), the coating material behaved very well, there being no detachment of the coating material from the base material;-Lack of coating detachment at the crack initiation and stable crack growth indicate very good bonding between coating and substrate;-Detachments of the coating material from the base material can be observed in the area of sudden crack propagation. There are pores or detachment, which is justified given the high values of deformations in this area and given the difference between the specific deformations (strains) of the two materials.

## Figures and Tables

**Figure 1 materials-15-03609-f001:**
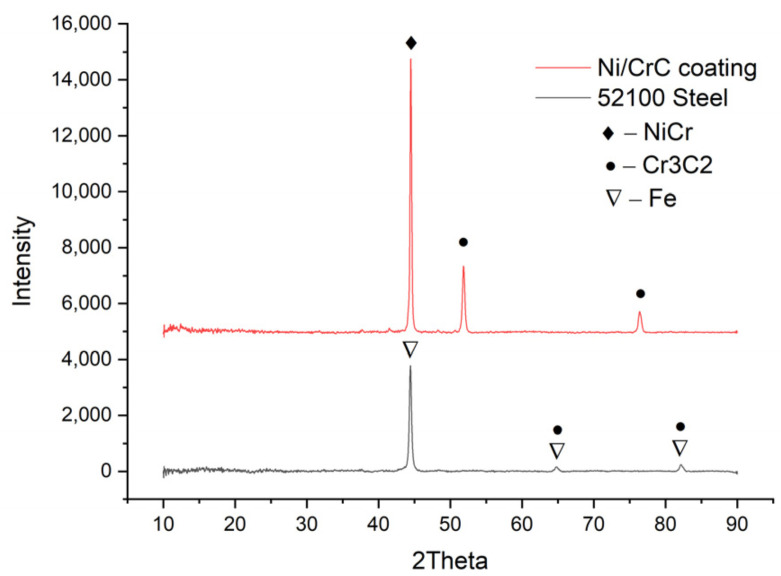
XRD pattern of coated sample and the base material.

**Figure 2 materials-15-03609-f002:**
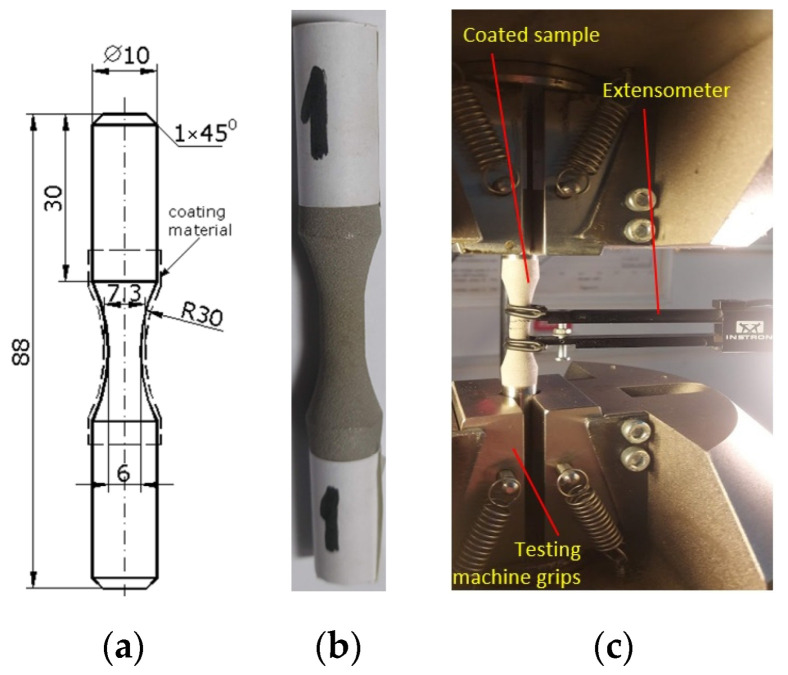
(**a**) The shape and dimensions of the sample (in mm). (**b**) Coated sample. (**c**) Attaching the sample to the testing machine.

**Figure 3 materials-15-03609-f003:**
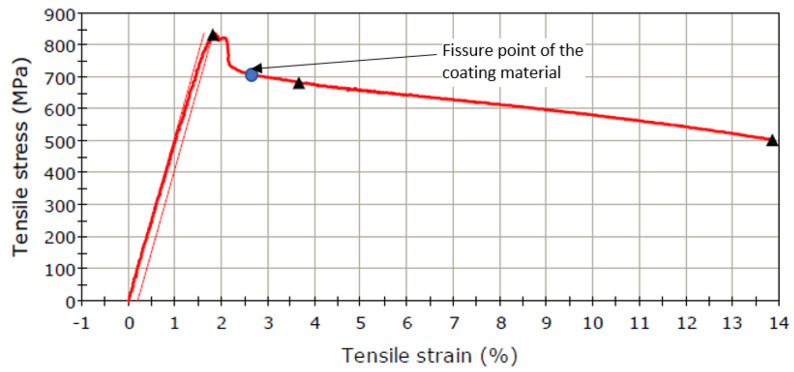
Characteristic stress-strain curve, specific to a coated sample.

**Figure 4 materials-15-03609-f004:**
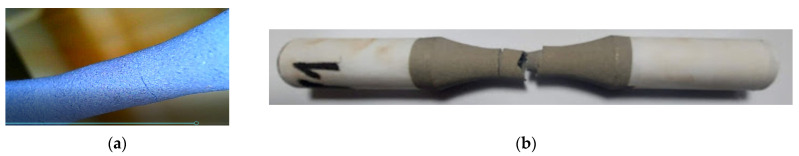
Sample subjected to static tensile load. (**a**) Image from the video that captures the moment of cracking of the coating material at the static tensile load; (**b**) broken sample after static tensile.

**Figure 5 materials-15-03609-f005:**
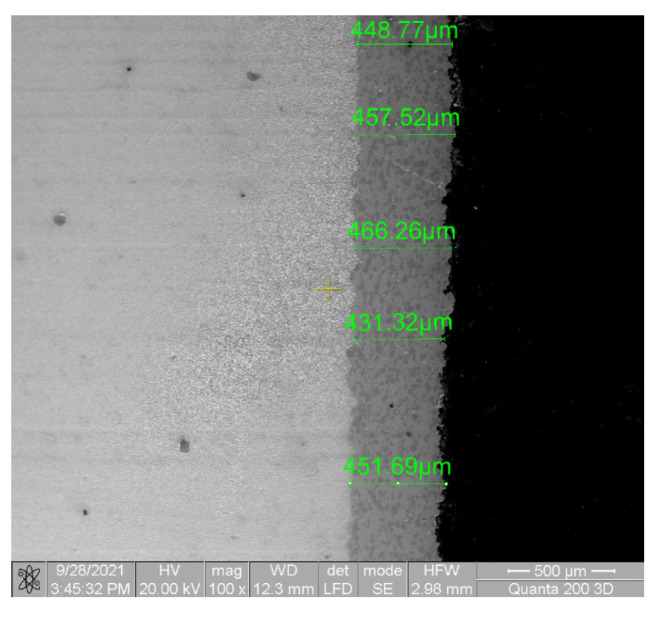
SEM image with cross-section at the interface between the base and the coated materials.

**Figure 6 materials-15-03609-f006:**
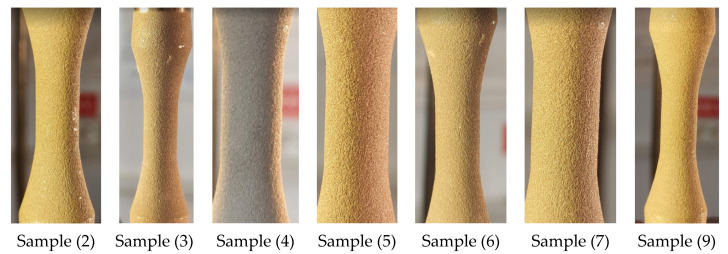
Outer surface during fatigue tests—different samples, near fracture point.

**Figure 7 materials-15-03609-f007:**
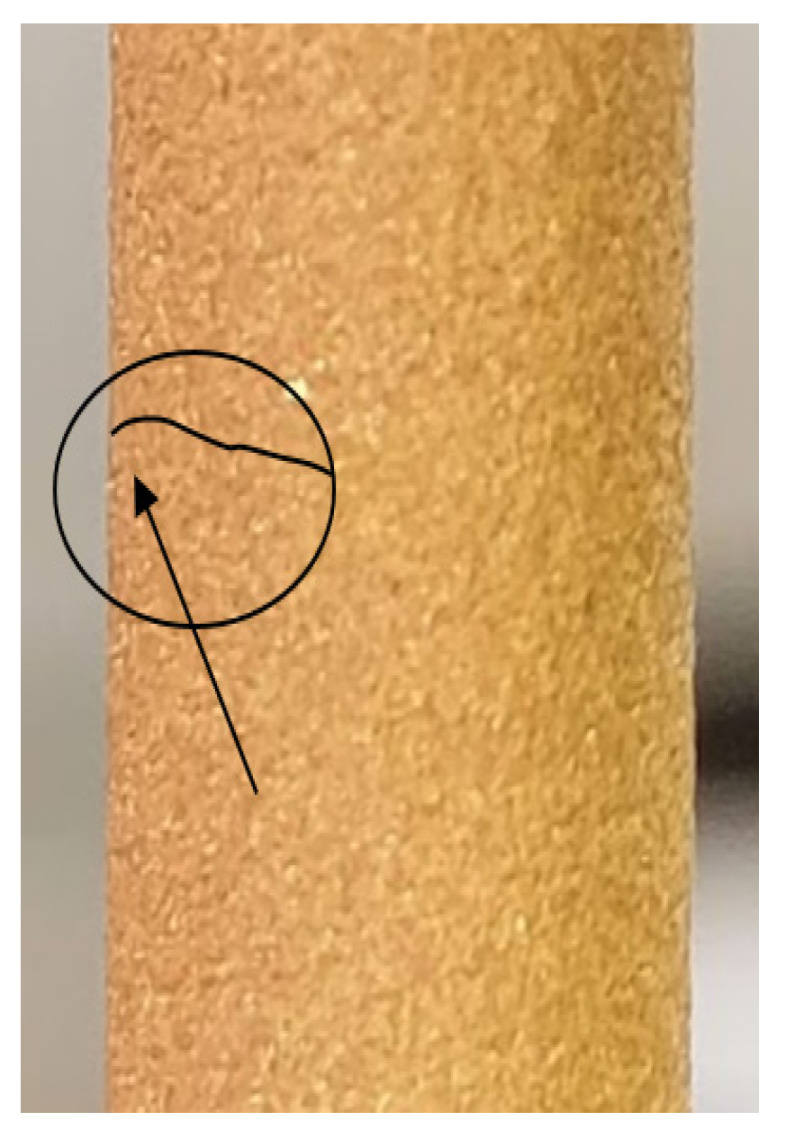
The appearance of fatigue cracks during fatigue loading.

**Figure 8 materials-15-03609-f008:**
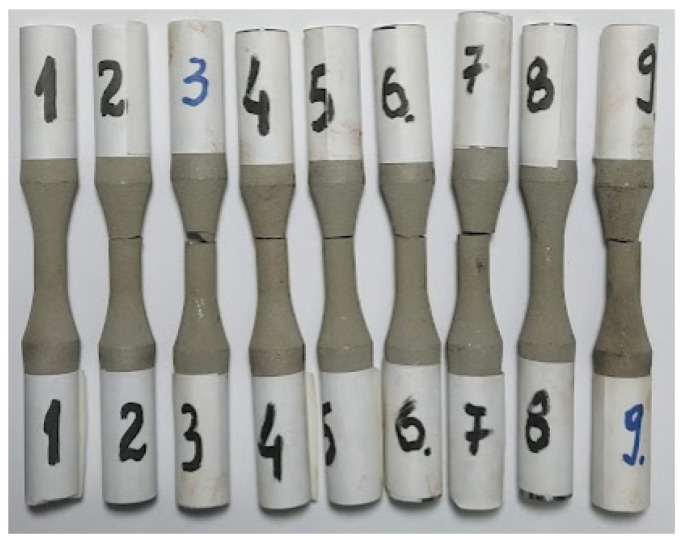
Samples after loading by alternate-symmetric fatigue tests.

**Figure 9 materials-15-03609-f009:**
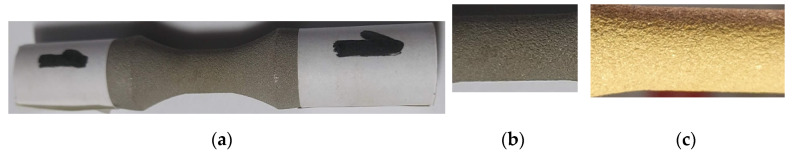
Macroscopic appearance of the outer surface after fatigue stress—Sample 1 σ_max_ = 420 MPa, N = 5,519,600 cycles. (**a**) the unbroken sample after more than 5 million cycles (8×). Macroscopic image of the surface for (**b**) 32× and (**c**) 45× magnification. Sample 8 had a similar appearance.

**Figure 10 materials-15-03609-f010:**
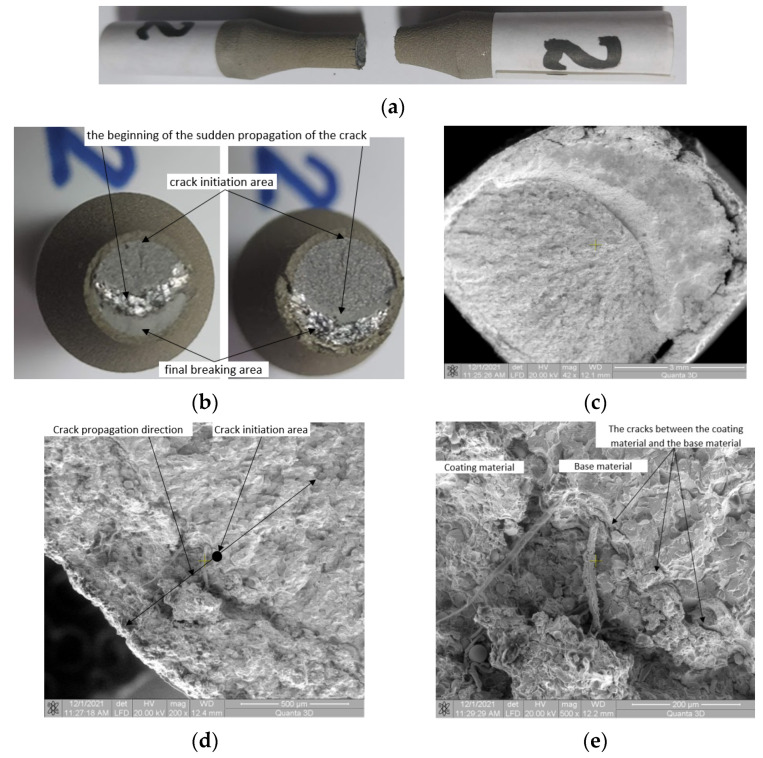
Macroscopic and microscopic appearance of broken surfaces for Sample 2, σ_max_ = 636 MPa, N = 5300 cycles. (**a**) Sample 2 broken by fatigue. (**b**) Fracture surfaces. (**c**) SEM image of the surface at 42× magnification. SEM image of crack initiation region with (**d**) 200× and (**e**) 500× magnification.

**Figure 11 materials-15-03609-f011:**
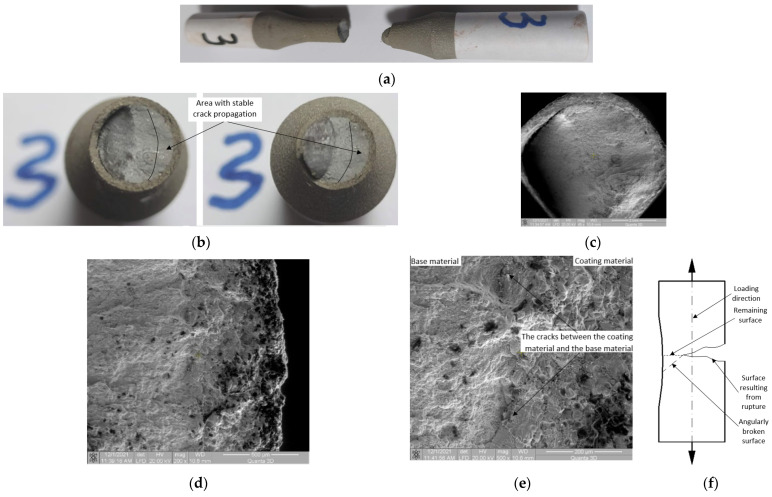
Macroscopic and microscopic appearance of broken surfaces for Sample 3, σ_max_ = 530 MPa, N = 34,660 cycles. (**a**) Sample 3 broken by fatigue. (**b**) Fracture surfaces. (**c**) SEM image of the surface at 45× magnification. SEM image of the area where the crack initiated with (**d**) 200× and (**e**) 500× magnification. (**f**) The remaining area before total failure.

**Figure 12 materials-15-03609-f012:**
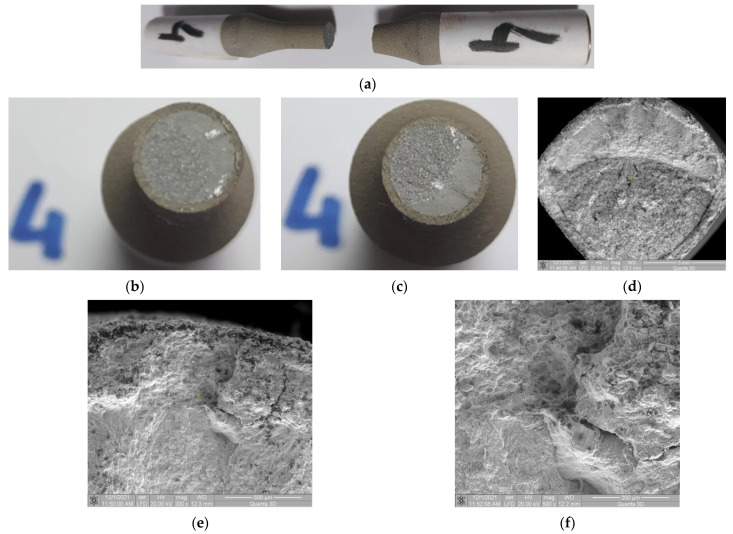
Macroscopic and microscopic appearance of broken surfaces for Sample 4, σ_max_ = 477 MPa, N = 878,965 cycles. (**a**) Sample 4 broken by fatigue. (**b**,**c**) Fracture surfaces. (**d**) SEM image of the fracture surface at 42× magnification. SEM image of the area where a crack initiated with (**e**,**f**) 200×, and 500× magnification.

**Figure 13 materials-15-03609-f013:**
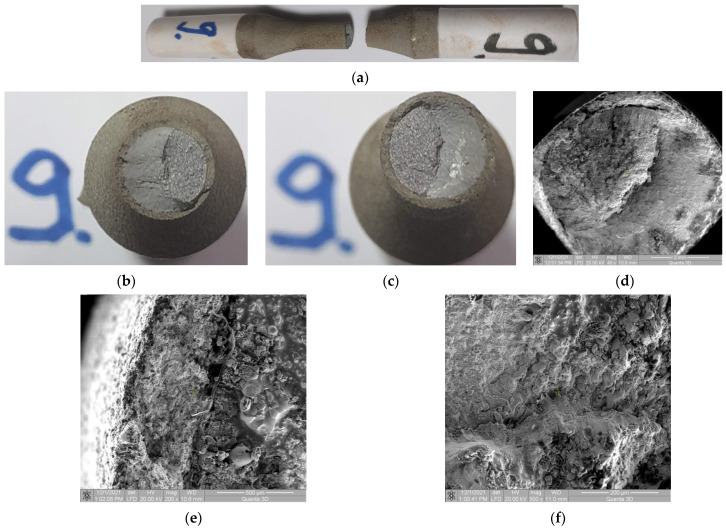
Macroscopic and microscopic appearance of broken surfaces for Sample 9, σ_max_ = 446 MPa, N = 4,424,120 cycles. (**a**) Sample 9 broken by fatigue. (**b**,**c**) Image of fractured surfaces. (**d**) SEM image of the fracture surface at 42× magnification. (**e**,**f**) SEM images of the area where a crack initiated with 200×, and 500× magnification.

**Figure 14 materials-15-03609-f014:**
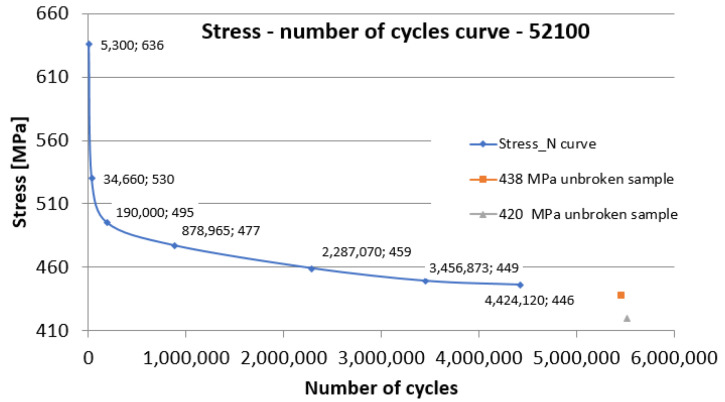
Wöhler curve for fatigue test of the 52100-steel coated with WIP-C1 powder.

**Table 1 materials-15-03609-t001:** Cold spray deposition parameters used to prepare the fatigue specimens.

Parameter	Value
Gas	Nitrogen
Pressure	6.2 MPa (900 psi)
Temperature	675 °C
Nozzle ID	WC NNZL0060
Nozzle throat size	2 mm
Powder feeder speed	8 rpm
Powder feeder gas glow	105 slm
Standoff distance	25 mm
Spray angle	90°
Nozzle traverse speed	250 mm/s
Nozzle step distance	0.5 mm
Layer thickness	0.127 mm
Target coating thickness	0.508 mm
Powder	WIP-C1
Bond coat	WIP-BC1 and 60°

**Table 2 materials-15-03609-t002:** Maximum stress σ_max_ and the number of cycles to failure N obtained for Ni/CrC coated 52100-steel samples.

Sample No.	σ_max_ (MPa)	N
2	636	5300
3	530	34,660
4	477	878,965
7	459	2,287,070
5	449	3,456,873
9	446	4,424,120
8	438	5,451,948
1	420	5,519,600

## Data Availability

Not applicable.

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
