# Peer review of "Evaluation of the Fatigue Behaviour and Failure Mechanisms of 52100 Steel Coated with WIP-C1 (Ni/CrC) by Cold Spray"

_materials, 2022, doi:10.3390/ma15103609_

Round 1
Reviewer 1 Report
Dear Authors,
The study is quite interesting technically. however a few points given below needs to be addressed for enhancing the quality of the manuscript.
- The language of abstract needs to be enhanced. In current form, it looks like a report.
- Introduction section lacks the purpose. Essentially, the role of fatigue strength in application.
- Fissure point shown in the figure needs to be explained.
- Barring sample 2 all other samples from the SEM looks like have filed in brittle mode. Do authors have a justification for same.
- Results and discussion section needs to be divided into sub sections.
- Have the authors carried out the chemical analysis of coated area?
- Conclusions section are mentioning new facts and claim. Those should be proved in previous section. Conclusion should just summarize the major take away.
Author Response
Response to Reviewer 1 Comments
Dear Reviewer,
Thank you very much for your comments. It certainly helped us to improve the manuscript. Please see below our answers for each point.
Point 1: The language of abstract needs to be enhanced. In current form, it looks like a report
Response 1: Thank you for the suggestion. We have updated the abstract text:
”Cold spray technique has been major improved in the last decades, for studying new properties for metals and alloys of aluminum, copper, nickel, titanium, as well as steels, stainless steels and other types of alloys. Cold sprayed Ni/CrC coatings have the potential to provide a barrier as well as improved protection to steels. Fatigue characteristics of 52100 steel coated with Ni/Chrome-Carbide (Ni/CrC) powder mixture by using cold gas dynamic spray are investigated. Fatigue samples were subjected to symmetrically alternating, axially applied cyclic fatigue loading until failure. The test was stopped if a sample survived more than 5x106 cycles at the applied stress. Fracture surfaces for each sample were examined to investigate the behavior of the coating both at high stress levels and at high number of stress cycles. Scanning electron microscopy was used to assess the damage in the interface of the two materials. Good fatigue behavior of the coating material was observed, espe-cially at low stresses and high number of cycles. Details of the crack initiation region, the stable crack propagation region and the sudden crack expansion region are identified for each sample. In most of the samples, the initiation of the crack occurred on the surface of the base material and propagated into the coating material. The possible effects of coatings on the initial deterioration of the base material and the reduction of the lifespan of the coated system was also investigated. The aim of the paper was to study the interface between the base material and the coating material at the fatigue analysis for different stresses, highlighting the appearance of crack and the number of breaking cycles required for each sample.”
Point 2: Introduction section lacks the purpose. Essentially, the role of fatigue strength in application.
Response 2: The specification below has been inserted in the Introduction, stating that the discussion on the comparison between the fatigue strength of the coated and the uncoated material is made at the end of Chapter 5, where a new specification is also introduced.
Introduction, at the end
“The paper also compared the fatigue resistance of 52100 steel given in reference [20] and the behaviour of the coating assembly presented in this paper, respectively, coating 52100 steel with a Ni/CrC mixture by using the cold spray technology.”
Chapter 5, at the end
“Tests performed by other researchers on the uncoated 52100 material, have shown that no limit to well-defined fatigue can be determined [20]. On the other hand, the base material, in this case 52100 steel, significantly influences the appearance of the S-N durability curve.”
Point 3: Fissure point shown in the figure needs to be explained.
Response 3: During the static test, we took a video capture of the sample, loading at traction. Looking at the movie, we saw exactly when the crack occurred. From the tables provided by the test machine we identified that moment and thus determined the exact point where the crack in the coating material was first visible. Below is a print-screen taken from the movie when you see the crack. The crack occurred at 7 sec. in a movie cut from the original movie that is longer.
Point 4: Barring sample 2 all other samples from the SEM looks like have filed in brittle mode. Do authors have a justification for same.
Response 4: The fatigue behaviour of the assembly is dictated by the behaviour of the 52100 steel. The coating material does not significantly influence this, except to the extent that the coating process causes damage to the surface of the base material. As a result, the fragile or ductile macroscopic appearance is given mainly by the base material. It is known that 52100 steel has a predominantly brittle breaking behaviour. Even sample 2 had a brittle behaviour because the breaking surface is perpendicular to the direction of application of the force and the cone-cup type behaviour, specific to ductile materials, is not observed.
Point 5: Results and discussion section needs to be divided into sub sections.
Response 5: Dear Reviewer, Thank you for your guidance and we appreciate your comments. We have adopted this style of writing, because we did the studies on each test sample, at breaking values and number of cycles. It is much easier to present these results as case studies for each sample, in which we highlighted the particularities, than to exemplify on 2 large chapters and to make references to other figures on different pages. It would require an arrangement with another numbering of figures, which means a new form of the paper. The other 2 reviewers did not mention anything in this regard to change the structure of the paper. Please take into consideration to keep this form due to the above mentioned facts.
Point 6: Have the authors carried out the chemical analysis of coated area?
Response 6: Dear Reviewer, thank you for your suggestion. We did not perform the chemical analysis, because we focused our attention on the fatigue study of the samples and the crack surface analysis at the interface between the base material and the coating. Also in the initial manuscript we introduced the reference on the chemical composition of the powder (ref. 18), exactly to highlight this aspect. At your suggestion and the availability of the existing infrastructure in this short time imposed by the Editor, we conducted XRD studies on the surface of the base material as well as on the coating in order to highlight their phases and correlate them with the chemical composition of the powder. We have introduced the description of the equipment as well as the figure and the specific comments in the text:
”The morphological aspect of the surfaces was determined using the SEM Quanta 200 3D (FEI , Czech Republic) electron microscope with LFD (Large Field Detector), High Voltage of 20 kV, spot size of 5 and a working distance of 15 mm. The X-Ray diffractions (XRD, Panalytical, Almelo, The Netherlands) were performed using a Xpert PRO MPD 3060 facility from Panalytical (Almelo, The Netherlands), with a Cu X-ray tube (Kα = 1.54051Å), 2 theta: 10◦–90◦, step size: 0.13, time/step: 51 s, and a scan speed of 0.065651◦/s.
Figure 1. XRD pattern of coating and base material
Figure 1 presents the diffractograms for the base and coating material. Knowing the chemical composition of the powder [18], the predominant phase of NiCr (cubic structure -ICDD 96-901-2971) was obtained at a 2Theta angle of approximately 44ᵒ and the secondary phase of Cr3C2 (cubic structure-ICDD 96-901-1599) at a 2Theta an-gle of approximately 53ᵒ and 77ᵒ. In the case of the basic material, the predominant phase is Fe (cubic structure -ICDD 96-901-6602), around the 2 Theta angle of 45ᵒ, and the secondary phases contain Fe and Cr3C2 at the 2Theta angles of 65ᵒ and 83ᵒ.
Point 7: Conclusions section are mentioning new facts and claim. Those should be proved in previous section. Conclusion should just summarize the major take away.
Response 7: In the “Conclusions” chapter, a series of statements made in “Results and discussion” are repeated. All the sentences lined up at the beginning have been said before, maybe in another form, putting them next to each other, to get an overview of the fatigue behaviour (cracking, damage, failure, detachment of material, fatigue strength) of the coated 52100 steel. There are several aspects that need to be highlighted in the conclusions, depending on the stress and the number of cycles to break. The fatigue behaviour, depending on these two parameters, was different, as a result no single conclusion could be drawn regarding the overall behavior of the coated 52100 steel.

Reviewer 2 Report
The manuscript is very clear and informative. However the following suggestions are made for further clarity.
- Citation style in introduction may be checked. for example in line 65. "This work? showed different opinions regarding...
- please include what is the criteria to select stresses in Table2, (636,530,477,459,.......420.). is it random or in some progression?
- why 9 samples only, any criteria to limit the study to 9 samples. and how can the results justified with 1 sample at each load. (to take care of experimental error?)
- In actual applications of these coatings, what is load condition? is it not exceeding 53%? (ref line 210)
- check for References between 2011 -2019, is this work not reported in that decade?
Author Response
Response to Reviewer 2 Comments:
Dear Reviewer,
Thank you very much for your comments. It certainly helped us to improve the manuscript. Please see below our answers for each point.
Point 1: Citation style in introduction may be checked. for example in line 65. "This work? showed different opinions regarding
Response 1: Dear reviewer, thank you for the observation. We have used the Journal’s template and respected the following text ” In the text, reference numbers should be placed in square brackets [ ], and placed before the punctuation; for example [1], [1–3] or [1,3]. For embedded citations in the text with pagination, use both parentheses and brackets to indicate the reference number and page numbers; for example [5] (p. 10). or [6] (pp. 101–105).” We don’t see any errors, but if there are the Editor will probably noticed us.
Point 2: please include what is the criteria to select stresses in Table2, (636,530,477,459,.......420.). is it random or in some progression?
Response 2: The first working stress must be less than the yield stress, approx. 0.8 of the yield stress. The following stresses must be decreasing. Those values came out because the testing machine requires forces that I gave with whole values. For example, the stress of 530 MPa was obtained as follows: σ = F/A = 15000 / 28.27 = 530 MPa, where the force was 15000 N and the cross-sectional area was 28.27 mm2. Stress values were further decreased, depending on what was obtained as the number of cycles at break.
Point 3: why 9 samples only, any criteria to limit the study to 9 samples. and how can the results justified with 1 sample at each load. (to take care of experimental error?)
Response 3: In the project (see Acknowledgments, Fundings) we were asked to study the fatigue behavior of the interface between the coating material and the base material and the overall fatigue behavior of the coated 52100 steel. It is true that, for a very precise determination of the fatigue strength for a limit of N0 cycles (which in this case was 5 million) it would have been possible to use several samples at each working voltage. We did not do this because of our goal. On the other hand, it is found that between the stress of 438 MPa (unbroken sample) and the stress of 447 MPa (broken test after approx. 4.4 million cycles) there is not a big difference and it was not productive to try other samples in this interval.
Point 4: In actual applications of these coatings, what is load condition? is it not exceeding 53%? (ref line 210)
Response 4: The question that arose in the project was: how high can you go with the working stress without the sample deteriorating after a number of 5 million cycles? The answer was that the stress of 438 MPa could not be exceeded. For coated steel 52100 this represented a resultant coefficient as the ratio between the test stress and the yield stress of 0.53. In general, for steels, this coefficient is between 0.45 and 0.55, but, as in our case it was a coated steel, it had to be determined experimentally, resulting in a value of 0.53.
Point 5: check for References between 2011 -2019, is this work not reported in that decade?
Response 5: Dear Reviewer, Thank you for your comment. We had no reason not to choose any reference from 2011-2019. The selection of the references was made exclusively on the basis of the closest possible comparison with the materials used, especially on the cold spray coating technology, as well as on the method of fatigue testing of the samples. At your recommendation, we have updated the list of references with titles from the mentioned period, but with other types of powders, respecting as much as possible the deposition technology and the test methods.

Reviewer 3 Report
This work is concerned on the fatigue study of one of steel materials coated with Ni/CrC by cold spray technique. The shape of the samples and study method are typical, with no invention in the studies. Anyway, results of this study can be useful, but the authors did not mention any application of these achievements. A comparison of the 52100 steel without the coat would be interesting, if e.g. no corrosion results are performed.
The manuscript is generally written with a good language, but there are some points to improve.
First of all, I wonder which „United States” are more important: these in lines 10 and 12 (>U.S.A.<) or those in lines 96, 123, etc. („USA”); or >>Surf Coat Technol<< in line 384 or „Surface and Coatings Technology” (lines 399, 401, etc.).
In your Table 1, which „degree” do you want to use, the first two >deg.<, or the last one (60⁰). The authors should unify their communicate. Line 36 (>Coatings must…<), in fact, „Coatings” do not have to do anything…, this is the role of „investigators”. For other remarks, see the PDF Enclosure with color highlights.
To sum up, the manuscript should be improved and corrected before it is considered for publication.
Moreover, the second conclusion could be expected without any additional studies, and here it is just a confir-mation of this fact. Such a mention could be added.

Author Response
Response to Reviewer 3 Comments
Dear Reviewer,
Thank you very much for your comments. It certainly helped us to improve the manuscript. Please see below our answers for each point.
Point 1: This work is concerned on the fatigue study of one of steel materials coated with Ni/CrC by cold spray technique. The shape of the samples and study method are typical, with no invention in the studies. Anyway, results of this study can be useful, but the authors did not mention any application of these achievements. A comparison of the 52100 steel without the coat would be interesting, if e.g. no corrosion results are performed.
Response 1: Dear Reviewer, Thanks for the feedback. Please note that these studies are part of the U.S.- Government contract, as it is written in the Acknowledgment, "This work was sponsored in part by Army Research Laboratories under grant number W911NF-20-2-0024." These studies are focused for mechanical, fatigue and wear analysis. Corrosion resistance research has not been the subject of this contract, but may be considered in the future as complementary studies. Due to the short response time for the review, it is impossible for us to carry out these studies, due to the need to approve this analysis by the project management and the long period of the analysis and interpretation of the results obtained.” A comparison of the 52100 steels without the coat would be interesting”:
We have updated the manuscript with the following information:
The specification below has been inserted in the Introduction, stating that the discussion on the comparison between the fatigue strength of the coated and the uncoated material is made at the end of Chapter 5, where a new specification is also introduced.
Introduction, at the end
“The paper also compared the fatigue resistance of 52100 steel given in reference [20] and the behaviour of the coating assembly presented in this paper, respectively, coating 52100 steel with a Ni/CrC mixture by using the cold spray technology.”
Chapter 5, at the end
“Tests performed by other researchers on the uncoated 52100 material, have shown that no limit to well-defined fatigue can be determined [20]. On the other hand, the base material, in this case 52100 steel, significantly influences the appearance of the S-N durability curve.”
The manuscript is generally written with a good language, but there are some points to improve.
Point 2: First of all, I wonder which „United States” are more important: these in lines 10 and 12 (>U.S.A.<) or those in lines 96, 123, etc. („USA”); or >>Surf Coat Technol<< in line 384 or „Surface and Coatings Technology” (lines 399, 401, etc.).
Response 2: Dear Reviewer, Thank you for reporting these errors. All are equally important. We have corrected the errors reported in the text.
Point 3: In your Table 1, which „degree” do you want to use, the first two >deg.<, or the last one (60⁰). The authors should unify their communicate. Line 36 (>Coatings must…<), in fact, „Coatings” do not have to do anything…, this is the role of „investigators”. For other remarks, see the PDF Enclosure with color highlights.
Response 3:
Dear Reviewer, thank you for reporting these errors. We have corrected the errors reported in the text and brought it to a unitary form.
To sum up, the manuscript should be improved and corrected before it is considered for publication.
Moreover, the second conclusion could be expected without any additional studies, and here it is just a confirmation of this fact. Such a mention could be added.

Round 2
Reviewer 1 Report
The manuscript has definite improvement over the previous version. It can now be accepted